# Inter-animal transforms as a guide to model-brain alignment

**Imran Thobani**
Department of Philosophyy
Stanford University
ithobani@stanford.edu

**Javier Sagastuy-Brena**
Wispr.ai
San Francisco, CA
jvrsgsty@stanford.edu

**Aran Nayebi**
McGovern Institute for Brain Research
Massachusetts Institute of Technology
anayebi@mit.edu

**Rosa Cao**
Department of Philosophy
Stanford University
rosacao@stanford.edu

**Daniel Yamins**
NeuroAILab
Stanford University
yamins@stanford.edu

## Abstract

The appropriate methods for aligning neural network models to the brain remain controversial. Ideally, a good alignment method should be powerful enough to enable accurate predictions of neural responses under a mapping from model units to neurons, while also being specific enough to distinguish the target system (e.g. a particular brain area) from other systems. It has generally been assumed that the goals of predictivity and specificity are in tension with each other, with methods that severely restrict the possible relationships between model and target being better for specificity, and more flexible methods yielding higher predictivity. We show that this apparent tension does not in fact exist. Fundamentally, this is because specificity requires not only distinguishing response patterns from different brain areas (i.e. separation), but also recognizing response patterns from the *same* brain area as being similar across subjects (i.e. identification). Taking this into account, we find that relatively flexible methods, like linear regression, can exhibit greater specificity compared to stricter methods, while also enabling better predictions. Motivated by the idea that the correct balance between strict and loose is manifested by the empirical relationships between subjects in a population, we introduce an alignment method that incorporates known aspects of the biological circuit, further improving predictivity without reducing specificity.

## 1 Introduction

Many deep learning models trained to perform cognitive tasks predict trial-averaged spiking responses accurately under linear regression (Yamins et al., 2014; Storrs et al., 2021; Kell et al., 2018; Zhuang et al., 2021). This raises the question of how to align model features to neural responses in order to assess the quality of neural networks as mechanistic models of the brain.

However, aligning neural networks to the brain has been challenging because it has been unclear what the criteria for good alignment methods are. Ideally, a good alignment method should succeed on two fronts. First, it should enable *accurate predictions* of neural activity, implemented via a mapping from model components to neural components that aligns simulated and real activity. Second, an alignment method should exhibit *specificity*: it should identify response patterns from the same part (e.g. brain area or model layer) as being similar across different instances of the population, while distinguishing response patterns from different parts as being dissimilar.

Predictivity is motivated on practical and philosophical grounds. A practical motivation is that we often want to use mechanistic models as stand-ins for real brains (or brain areas) by accurately simulating brain responses. In order for mechanistic models to be useful as stand-ins, high mechanistic similarity should go hand-in-hand with high predictive accuracy with respect to brain responses. In addition, recent philosophical work on mechanistic models in neuroscience argues that a good mechanistic model of a system should have variables that can be accurately mapped to some set of variables in the target system that are causally sufficient to generate the behavior of interest (Cao & Yamins, 2021). More specifically, we should be able to transform one *runnable model or system* into another under an appropriate mapping. In what follows, our working assumption is that firing rates are causally sufficient to generate the behaviors we want to explain, and we therefore assess similarity by mapping between model responses and firing rates in a population of animals.

The motivation for specificity is that, in order to be a *mechanistic* mapping, the model-brain mapping should identify similarity between response patterns of the *same* type while distinguishing response patterns of *different* types. For example, if an alignment method mapped retinal responses onto IT responses with high accuracy, then it would be eliding important functional distinctions between brain areas that are performing very different operations. At the same time, the alignment method should be able to identify genuine functional similarities between response patterns from the *same* brain area, even when they come from different subjects and therefore are not *exactly* identical.

It has been widely presumed that the goals of predictivity and specificity are in tension with each other Ivanova et al. (2021). Intuitively, more flexible transform classes appear better for prediction, while stricter transform classes appear better for separation. To the extent that this trade-off exists, there is an inherent divergence between the goals of accurate prediction (e.g. building brain-machine interfaces) and scientific understanding (e.g. systems identification). Indeed, this assumption has had substantial influence on metric design, with researchers pursuing scientific understanding favoring stricter methods Williams et al. (2021) and those pursuing engineering applications favoring more flexible methods Schrimpf et al. (2018).

However, the literature overlooks a crucial aspect of specificity: identifying similarity between response patterns of the same type (e.g. from the same brain area or model layer) (Fig. 1B). Indeed, an alignment method that indiscriminately separated *all* response patterns would be incapable of recognizing target systems of the same type (e.g. the same brain area and species) as similar to each other, and therefore would lack specificity. Considering both aspects of specificity suggests that rather than a trade-off, there is an optimal balance between strictness and flexibility, where we want the *narrowest* class of transforms that accurately maps responses between subjects for the same brain area (Fig. 1C). Once this transform class has been identified, it is then possible to measure model-brain similarly using this same transform class, in effect computing how well an ANN can masquerade as an element of the real population using the same "rules of similarity" as needed to compare real animals (Fig. 1D). To better approximate the ideal transform class, we propose a transform class that accounts for known aspects of the biological circuit, increasing predictivity without reducing specificity.

This framework helps address recent debates about how strict or flexible a similarity score should be. For example, many researchers (Kornblith et al., 2019; Ding et al., 2021; Conwell et al., 2022; Finzi et al., 2022) have pushed for stricter similarity assessments than linear regression. One recent example is Khosla & Williams (2023)'s soft matching method, which assesses similarity with respect to tuning curves of individual neurons, while also being able to handle neuronal populations of different sizes and satisfying criteria for a mathematical metric, following prior work (Williams et al., 2021). Our approach requires a good transform class to map accurately across subjects for the same brain area (while still distinguishing different brain areas), and a transform class that fails to do so will therefore be *too* strict. Moreover, it may turn out that a good transform class leads to similarity scores that do *not* satisfy metric criteria.

We find that a good transform class must take into account aspects of the Linear-Nonlinear structure that occurs in biological (and artificial) neural networks. To see why, consider that even if linear filter outputs at each layer (*before* applying the non-linear activation function) are linearly related between subjects, this does not imply that firing rates (which are *post*-non-linearity) are so related. The non-linear activation function may well distort the inter-subject transforms. Indeed, when we look at a population of DNN models of mice, we find that post-non-linearity responses in intermediate layers are not very similar according to a linear transform, but pre-non-linearity responses are. By

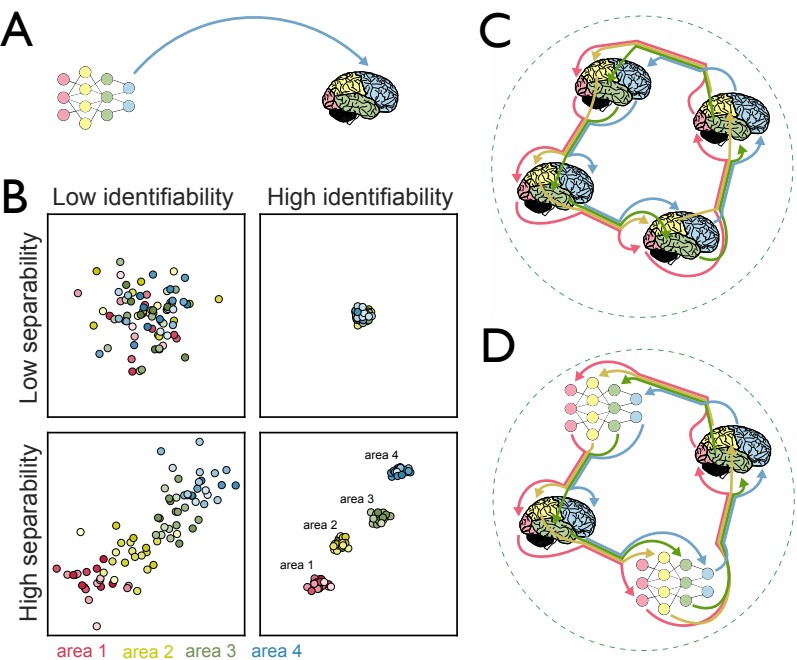

Figure 1: Inter-animal transforms as a framework for model-brain comparison (A) Predictivity: An alignment method should provide a mapping from model units to neurons, so that a good model can predict neural activity for scientific understanding or engineering applications (e.g. brain-computer interfaces). (B) Specificity: An alignment method should also exhibit specificity, meaning that it separates response patterns of different types (high separability), such as responses from different brain areas, while also identifying similarity between response patterns of the same type (high identifiability), such as responses from the same brain area. Identifiability is just as critical as separability to specificity, though it is often overlooked. (C) A natural way to achieve both predictivity and specificity is to identify the *narrowest* class of transforms that maps responses between subjects for a given brain area and species. (D) Once identified, the inter-animal transform class can be used to map a candidate model's responses in a given layer to a given brain area in order to assess how well the model responses can masquerade as responses of a typical animal subject.

analogy to the models, we hypothesize that the linear-nonlinear structure of the *biological* network also modifies the inter-animal transform class.

To account for details of the biological mechanism, we propose a transform class, Linear Nonlinear, that approximately inverts the biological non-linearity, applies a linear transform, and then re-applies a non-linearity (exponential) to predict the target neuron's firing rates. On an electrophysiological dataset of 31 mouse subjects, Linear Nonlinear increases same-area similarity scores relative to ridge regression and soft matching, while maintaining inter-area separability. Finally, we assess model-brain similarity for several models of the mouse visual system under different transform classes. We find that Linear Nonlinear and Ridge are just as good at separating models as Soft Matching, but only when both directions of fit (model to brain *and* brain to model) are taken into account. Thus, as with the models, there is no systematic tradeoff between predictivity and specificity.

Section 2 details our results in mapping between instances in a simulated population of DNN models of the mouse visual system to show how the form of the mechanism constrains the inter-model (and, we hypothesize, inter-animal) transform class. In section 3, we develop our biologically motivated transform class and show that Linear Nonlinear improves both predictivity and specificity on the model population. Finally, section 4 investigates how well our results on the model population generalize to a real mouse population.

## 2   A MODEL-BASED APPROACH TO DEVELOPING AN IMPROVED TRANSFORM CLASS

To generate hypotheses for a good transform class, we do model-model mappings in a simulated population of subjects using a state-of-the-art model of mouse visual cortex (Nayebi et al., 2022) in order to observe how well candidate transform classes map across model instances for the same layer. As we will see, the model-model mappings yield insights about how the form of the mechanism (perfectly observable in the case of the models) constrains a good transform class. Another benefit of using models is that we can observe responses for every unit and arbitrarily many stimuli (unlike the electrophysiological data) and therefore get more accurate similarity scores according to each candidate transform class.

Our models are based on a modified AlexNet architecture developed by Nayebi et al. (2022), which was found to predict mouse visual responses better than other models using linear regression. This model has relatively low-resolution 64x64 inputs and is trained on an unsupervised objective (Instance Discrimination) over the ImageNet training set. We further modified these models to use a softplus activation function (App. C) followed by Poisson-like noise (App. D) to better mimic neuronal response characteristics. To generate the population of "conspecific" models, we vary the random seed controlling the weight initialization and training data order. Each model subject is trained to have equally good performance as the original model on both the contrastive objective and on transfer performance on ImageNet object categorization (which it was not explicitly trained on). We hypothesize that varying the random seed leads to model response variability for a given layer that is approximately similar to variability in animal responses for a given brain area. To the extent that this approximation holds, a good inter-model transform class should also be a good inter-animal transform class, and the ranking of different transform classes (in terms of how accurately they map across subjects) should be roughly similar for both the simulated model population and the mouse population. As we will see in Sec. 4 (compare Fig. 3 and Fig. 4), this is the case.

### 2.1   PRE-NONLINEARITY ACTIVATIONS ARE HIGHLY SIMILAR UNDER A LINEAR TRANSFORM, LEADING TO A HIERARCHICAL TRANSFORM CLASS

With our model population, we now ask how well responses for different model subjects predict each other for the same layer using ridge regression. We see that post-softplus responses look fairly dissimilar at intermediate layers between models according to ridge regression (Fig. 2A). This is a striking result given that ridge regression has been criticized as too flexible. If model instances are not similar even according to ridge regression at the intermediate layers, one might suspect that the relations between different model instances are highly non-linear, involving (for instance) a sequence of non-linear operations similar to those in a deep neural network. If so, then different model instances perform very *different* operations in intermediate layers, even though we see that the first and last layers are somewhat similar, presumably because they are closer to the inputs and outputs, which are the same across model instances. This explanation sounds plausible, since it isn't obvious that differently seeded models *must* use similar operations to transform identical inputs to identical outputs, even for the same architecture (although they *could*, for all we know).

However, this explanation turns out to be wrong. As it turns out, pre-softplus responses *are* highly similar under a linear transform (Fig. 2A). The softplus non-linearity interferes with the response similarity according to linear transform at each layer of the model. This result reveals a non-obvious convergence between differently seeded model instances at intermediate layers. If no such convergence had occurred, then it would be unclear whether or not different model instances have functionally similar response patterns, and in turn, whether different *mouse subjects* have similar response patterns.

These results suggest that response similarity is most evident when assessed pre-nonlinearity and under a linear regression. The implication for mapping models to animals (or animals to each other) is that we should use linear regression to map *pre*-non-linearity responses to each other instead of *post*-non-linearity responses. Therefore, to compare models to brains, it may be useful to collect EPSP data in the future instead of spiking activities (Fig. 2B). However, given that EPSPs are currently difficult to measure, it is useful to develop a good transform class for post-non-linearity responses.

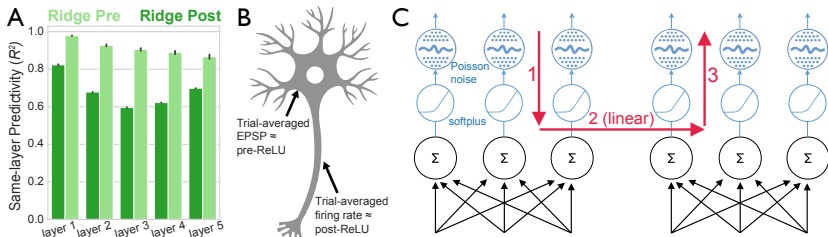

Figure 2: **A good transform class should take into account the neuronal activation function.** (A) When ridge regression is applied to post-softplus responses (dark green), same-layer predictivity is somewhat low between different model subjects. However, same-layer predictivity is much higher for ridge regression when evaluated pre-softplus (light green). (B) Pre-softplus responses can be thought of as corresponding to trial-averaged EPSPs in a biological neuron, while post-softplus responses correspond to trial-averaged firing rates. Since EPSPs are hard to measure, we want an alignment method that works well for post-non-linearity responses. (C) Taking into account the non-linearity requires a transform class with a hierarchical structure. Step 1 inverts the non-linearity in order to recover the pre-non-linearity activations of one model, step 2 applies a fitted linear transform to predict the pre-non-linearity activations of the other model, and step 3 re-applies the non-linearity to predict the post-non-linearity activations of the other model.

## 3 A BIOLOGICALLY MOTIVATED TRANSFORM CLASS

Given that pre-non-linearity responses are highly similar under a linear transform, a natural way to map post-non-linearity responses between two sets of neurons would be to, first, *invert* the non-linearity to recover the pre-non-linearity responses of the source neurons; second, apply a linear map to predict the pre-non-linearity responses of the target neurons; and finally, re-apply the non-linearity to predict the post-non-linearity responses of the target neurons (Fig. 2C).

For real firing rate data, we do not generally know the *exact* form of the activation function for each neuron, so we started with a transform class that only approximately captures the activation function. Linear Nonlinear approximately inverts the activation function using Yeo-Johnson scaling (Appendix E) and then applies a linear transform followed by the exponential function to approximate the smooth activation function in the models (see Appendix F for details). As with ridge regression, we use a cross-validated ridge penalty on the weights of the linear mapping to reduce over-fitting.

We also investigated what would happen if we did account more precisely for the activation function by considering two variants of Linear Nonlinear. Linear Softplus applies the softplus activation function in the last step of the hierarchical transform (Fig. 2C) instead of the exponential. Pre Linear Softplus goes one step further, mapping *pre*-non-linearity responses of one model subject to post-non-linearity responses of the other, which amounts to exactly (instead of just approximately) inverting the activation function in the first step. These progressive improvements lead to greater predictivity. Thus, accounting for aspects of the biological mechanism seems to be crucial for maximizing the predictivity of the transform class.

Finally, to compare these relatively flexible methods to a much stricter method, we evaluated soft matching, which assesses tuning curve similarity of individual neurons between two neural populations. It turns out that soft matching can be formulated as a predictive mapping (see App. B), and therefore can be evaluated for predictivity. We find that soft matching predictivity is low at all layers, especially intermediate layers (Fig. 3A). Therefore, soft matching is too strict to characterize the inter-model transform class.

Although our biologically motivated transform class (and its variants) performs better on predictivity, a crucial question is whether this improvement comes at the cost of specificity. In fact, given that we are now using *non-linear* transform classes (and non-linear transform classes are often thought of as being very flexible), this is a natural worry to have. We therefore evaluate all of the above methods for specificity.

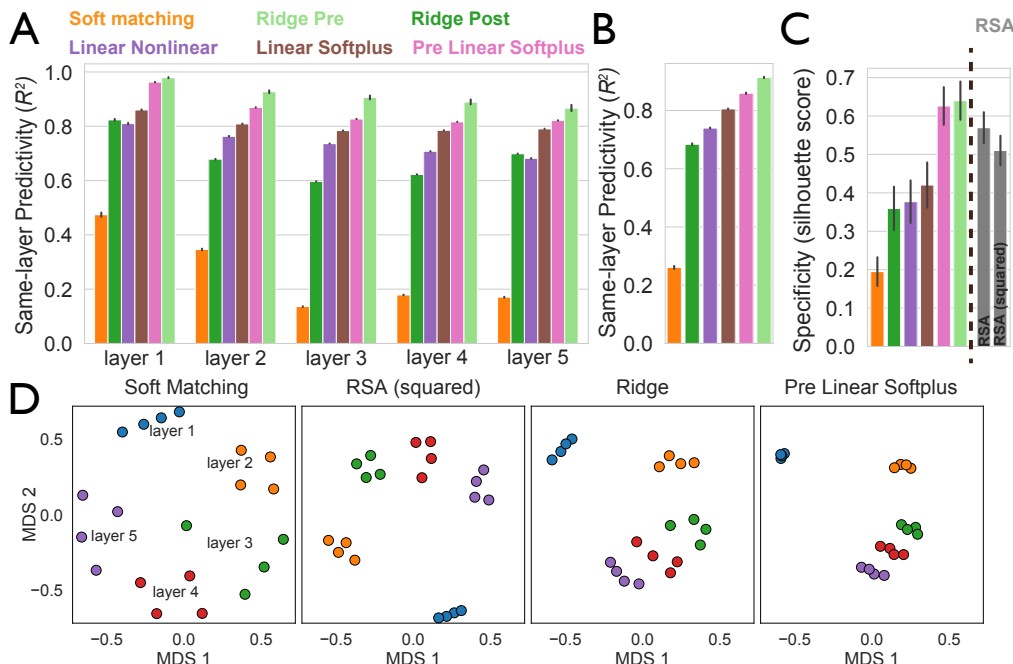

Figure 3: Results on simulated population. (A) Same-layer predictivity on model subjects. Accounting for aspects of the biological mechanism, specifically the activation function, improves predictivity. Pre Linear Softplus, which exactly accounts for the softplus activation function, best predicts post-softplus responses. The remaining gap between Pre Linear Softplus and Ridge Pre can be attributed to the fact that Ridge Pre predicts *pre-softplus* responses whereas Pre Linear Softplus predicts *post-softplus* responses, and so the $R^2$ score (which depends on the total variance of the target variable) can be different in the two cases. (B and C) Overall same-layer predictivity vs specificity. We do not see a systematic tradeoff between predictivity and specificity. In fact, the most predictive methods have the greatest specificity. RSA, which does not enable firing rate predictions, does have high specificity, but not as high as Pre Linear Softplus. (D) To understand *why* improving predictivity can improve specificity, we can visualize the assessed dissimilarities between response profiles using multi-dimensional scaling. The distances between points in the 2D plot are optimized to be as close as possible to the assessed dissimilarites between response profiles. The MDS plot reveals that methods with higher predictivity improve *identifiability*, a crucial component of specificity, while largely maintaining inter-layer separability. Of course, if separability had been lost, then the increased predictivity would not have resulted in increased specificity (cf. Fig. 1B).

To take into account *both* aspects of specificity (identifiability and separability), we compute the silhouette score Rousseeuw (1987), which is close to 1 just in case responses for different layers are separated much more than responses for the same layer. The silhouette score for response profile $i$ is:

$$s(i) = \frac{b(i) - a(i)}{\max(b(i), a(i))}$$

where $a(i)$ is the mean dissimilarity between $i$ and other response profiles for the same model layer, and $b(i)$ is the mean dissimilarity between $i$ and responses from the next most similar model layer. We compute the mean silhouette score over all model subjects and layers.

Rather than finding a trade-off, we find that increased predictivity can *improve* specificity (Fig. 3B, C) as long as inter-layer separation is maintained. For example, ridge (post-softplus) exhibits *more* specificity than soft matching (Fig. 3B). This is because ridge clusters same-layer responses more tightly than soft matching, thus improving *identifiability*, a key component of specificity, while maintaining inter-layer separation (Fig. 3D). Pre Linear Softplus achieves an even higher silhouette score, again because higher predictivity lead to higher identifiability. In fact, a maximally specific

transform class should achieve maximum predictivity for same-layer responses (to improve identifiability), while being as narrowly defined as possible (to improve or at least maintain separability).

It is worth stressing that, while more predictive methods *can* exhibit greater specificity, this is not always the case. For example, an extremely flexible alignment method, such as a deep multi-layer perception, could in principle map between many pairs of response profiles with high accuracy, regardless of whether they came from different brain areas or the same brain area. This would be a situation where we had high predictivity but low specificity (because of a lack of separability). One reason separability was maintained for our biological transform class is that it is *not* an unconstrained alignment method. Indeed, a core contribution of our approach is that the correct inter-animal transform class must take into account aspects of the biological mechanism and is therefore highly constrained.

## 4 Evaluating transform classes on mouse electrophysiology data

We investigate how well our results generalize to a mouse dataset containing Neuropixels recordings for 31 subjects in response to 118 naturalistic stimuli, averaged over 50 trials (de Vries et al., 2020). With only about 50 neurons measured per subject and brain area, we pool N-1 subjects' neurons to evaluate same-area predictivity for a target subject. Overall, the rank ordering of alignment methods in terms of same-area predictivity is similar for the real population as for the simulated population (Fig. 4A). This helps validate our simulated population as a model of inter-animal variability, at least to some degree.

The fact that Linear Softplus does not seem to perform significantly better than Linear Nonlinear means that we cannot tell, based on the transform class performance, that softplus is a better approximation to the activation functions in play than the exponential. However, it is worth noting that Linear Nonlinear is easier to fit, as it does not require an extra scaling parameter as Linear Softplus does. This means that on a limited number of stimuli, Linear Nonlinear may achieve strong performance, even if the exponential function is less similar to biological activation functions than softplus.

With predictivity scores in hand, we also want to evaluate specificity on the mouse data. When pooling across subjects, we cannot compute silhouette scores. We therefore assess specificity *indirectly* by considering the average difference between 4 candidate models in terms of assessed similarity to each brain area. A method with low specificity would not be able to differentiate models that are *more* similar to the brain from those that are *less* similar, and therefore would have low model separability. We consider four different models: the ReLU-based AlexNet model of mouse visual cortex (Nayebi et al., 2022), our noisy softplus version of that model, a ResNet model trained on ImageNet categorization with 64x64 resolution stimuli, and a VGG-16 model trained on ImageNet categorization, with 224x224 resolution stimuli (unlike the relatively low resolution mouse visual system).

We map model-to-brain as well as brain-to-model. For model-to-brain, soft matching separates models better, consistent with Khosla & Williams (2023), but for brain-to-model, Ridge and Linear Nonlinear separate models better (Fig. 4B). A possible reason is that model responses can have patterns not present in the brain data, and flexible mappings like Ridge or Linear Nonlinear may only detect such a discrepancy when mapping from brain to model. When mapping in both directions, model separability is as good for Ridge and Linear Nonlinear as it is for soft matching. Overall, there is not a trade-off between predictivity and model separability (Fig. 4C, D).

## 5 Conclusion

There is not a systematic trade-off between predictivity and specificity. In fact, both goals should be achieved by the *narrowest* class of transforms under which subjects' responses predict each other with high accuracy for that area. To better approximate that class, we introduce a method that accounts for the activation function, improving predictivity while maintaining specificity. Because the ideal inter-animal transform class should be as narrowly defined as possible to improve separability (subject to the requirement of maximizing predictivity), future research should investigate whether

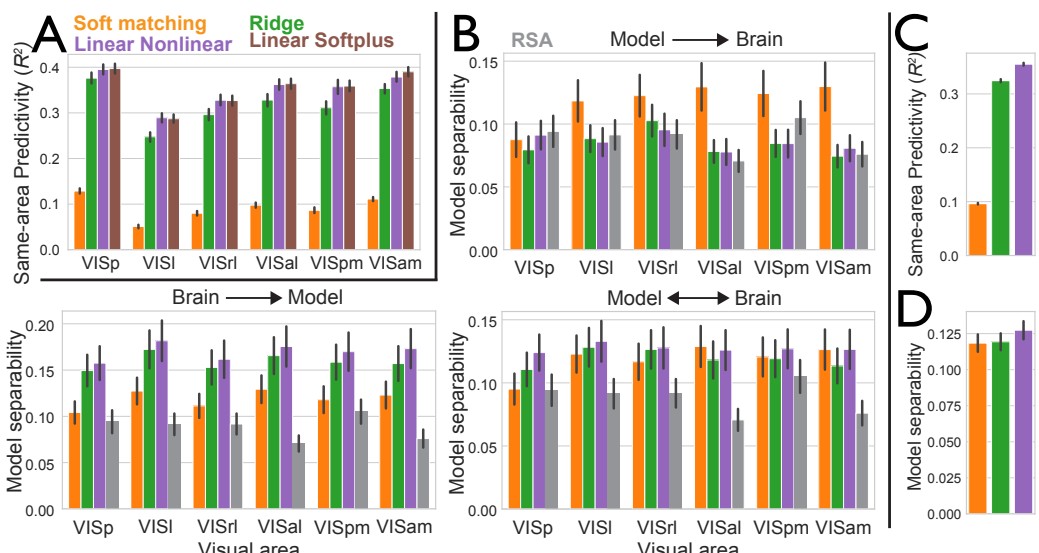

Figure 4: Results on mouse electrophysiology data. (A) Same-area predictivity when pooling neurons over N-1 subjects to predict responses in a held-out subject. The ordering of transform classes in terms of predictivity is roughly consistent with that observed in the simulated population (cf. Fig. 3A). (B) Mean separation between different candidate models in terms of their assessed brain similarity. We consider four candidate models: the ReLU based AlexNet model of mouse cortex by Nayebi et al. (2022), our noisy softplus version of that model, ResNet, and VGG16. We map models to brains, brains to models, and also do both directions (averaging the similarity scores over both directions before computing the separation between models). (C and D) Overall same-layer predictivity vs model separability. We do not see a systematic tradeoff between same-layer predictivity and model separability.

we can further constrain Linear Nonlinear (or its variants) in a way that improves specificity without reducing predictivity.

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

## A    CENTRAL HYPERCOLUMN SELECTION

In order to map between units with similar functional roles (at least for the same model layer), we do our model-model fits using only the central hyper-column of units in each layer (i.e. the units whose receptive field is directly at the middle of the input image). Indeed, even when constraining the mapping to use only the central hyper-column, we are able to identify high similarity across model instances for the same layer, at least when assessing pre-non-linearity responses using a linear transform.

## B    SOFT MATCHING AS A TRANSFORM CLASS

While Khosla & Williams (2023) do not explicitly formulate the soft matching score as a predictive mapping, it can be formulated as one. Computing the soft matching score involves maximizing:

$$\Sigma_{i,j}\mathbf{T}_{ij}\mathbf{C}_{ij}$$

where $\mathbf{T}$ is the transport matrix, subject to the constraints that the columns of the matrix sum to $1/N_Y$, while the rows sum to $1/N_X$, and $\mathbf{C}$ is the matrix of Pearson correlations between each source neuron and each target neuron. The transport matrix can be interpreted as a joint probability distribution over source neurons and target neurons (where the marginal distributions are uniform

discrete). Thus, the soft matching score is the *expected* correlation between source and target neurons, according to joint probabilities encoded by the optimal transport matrix.

Since maximizing the above objective requires identifying source neurons that are highly correlated with each target neuron, we can use the source neurons to predict the value of each target neuron, weighted by the probabilities in the optimal transport matrix. First, for each source neuron $X_i$ and target neuron $Y_j$, we can predict $Y_j$'s responses across a set of stimuli (symbolized as the vector $\mathbf{Y}_j$) based on $X_i$'s responses to those stimuli (symbolized as $\mathbf{X}_j$) as:

$$\hat{\mathbf{Y}}_j = \frac{\sigma(\mathbf{Y}_j)}{\sigma(\mathbf{X}_i)}[\mathbf{X}_i - \bar{\mathbf{X}}_i]\mathbf{C}_{ij} + \bar{\mathbf{Y}}_j$$

This is essentially using the correlation $\mathbf{C}_{ij}$ to do ordinary least squares between $X_i$'s responses and $Y_j$'s responses.

For a single target neuron $Y_j$, we compute the expected value of these correlation-based predictions across source neurons, if we sampled source neurons according to the conditional probability distribution $P(X = X_i | Y = Y_j)$. Since $\mathbf{T}_{ij} = P(X_i, Y_j)$ and $P(Y_j) = 1/N_Y$, it follows that $P(X = X_i | Y = Y_j) = N_Y \mathbf{T}_{ij}$. Using these conditional probabilities, the overall prediction $\hat{\mathbf{Y}}_j$ then becomes:

$$\hat{\mathbf{Y}}_j = N_Y \sigma(\mathbf{Y}_j) \Sigma_i \frac{\mathbf{X}_i - \bar{\mathbf{X}}_i}{\sigma(\mathbf{X}_i)} \mathbf{T}_{ij} \mathbf{C}_{ij} + \bar{\mathbf{Y}}_j$$

## C  MOTIVATING THE SOFTPLUS ACTIVATION FUNCTION WITH A SIMPLE MODEL OF A NOISY SPIKING PROCESS

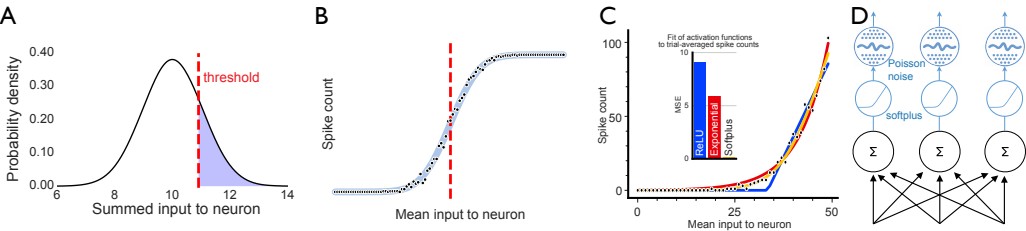

Figure 5:  A more biologically consistent activation function. (A) Biological activation functions are the result of a noisy spiking process. Because summed inputs to neurons are noisy, the firing probability is positive even when the mean input is sub-threshold. Here, the probability of spiking is represented as the size of the blue region. (B) The resulting activation function, unlike ReLU, is strictly positive and increasing. Dots represent simulated spike counts, which are Poisson-distributed in the limit of very small firing rates. (C) Fitting different activations to simulated spike counts, allowing for scaling and translation. Softplus fits spike counts the best in the sub-threshold regime. The exponential activation also function performs somewhat better than ReLU. Intuitively, the reason ReLU does not fit as well is that it has a hinge that prevents it from capturing the smooth increase in firing rate. Spike counts are plotted for a single trial. (D) We replaced each ReLU non-linearity in the models with a softplus non-linearity and a Poisson-like noise sampler.

Our simulation of spike counts is based on the following highly simplified model. We assume that a neuron receives total input $X \sim \mathcal{N}(\mu, \sigma^2)$ and that during a single time interval equal to the neuron's refractory period, the neuron either fires once or not at all, depending on whether $X > T$, where $T$ is a fixed threshold. We count the number of spikes over a 100 ms time range, and average over 100 trials.

Under this model, the total mean (over trials) spike count $S_t(\mu)$ over a time period $t$ (expressed as a function of the mean total input to the neuron $\mu$) is equal to $t/R * \Phi(\mu - T, \sigma^2)$, where $\Phi$ is the Gaussian CDF. This means that the activation function should have a sigmoid shape, which saturates at sufficiently high mean inputs. However, many cortical neurons are thought to fire in the fluctuation driven, unsaturated regime (Van Vreeswijk & Sompolinsky, 1996). We therefore focus on unsaturating functions like softplus and fit these functions to spike counts that we simulated in the unsaturated regime.

## D    NOISY SOFTPLUS ALEXNET MODELS

To obtain our noisy softplus variant of the AlexNet mouse model, every ReLU sub-layer in the AlexNet models is exchanged for a Softplus sub-layer followed by a Poisson-like noise block whose mean is the output of the Softplus sub-layer. PyTorch enables noisy models to be trained using a reparameterization trick, but only for certain probability distributions (not for the Poisson distribution). We use the Gamma distribution as a stand-in for Poisson, choosing shape parameter $k = \lambda$, where $\lambda$ is the Poisson parameter (which is chosen to be the output of the Softplus sub-layer), and scale parameter $\theta = 1$. This allows us to replicate two statistical properties of Poisson variables: non-negative samples, and variance-mean ratio of 1, both of which are important for using the Linear Nonlinear transform (which uses a Poisson GLM) to predict the responses. To avoid numerical difficulties for small values of $k = \lambda$, we scale the softplus outputs by 100 before sampling from the Gamma distribution. We then train the noisy softplus models so that their instance recognition training score (as well as validation score on ImageNet categorization) are equal to those of the ReLU-based AlexNet models.

## E    YEO-JOHNSON SCALING IN LINEAR NONLINEAR AND ILSP

When mapping models to models or models to brains, we can just use the pre-non-linearity features of the source model to assess response similarity. However, when mapping animals to animals (or, if enough neurons are measured, animals to models), we cannot easily obtain EPSP data, nor can we easily invert the activation function if we do not know its exact form for a given neuron. Yeo-Johnson scaling uses a power transformation to make the features closer to normally distributed over the stimuli. We expect this transformation to make the post-non-linearity features more correlated with pre-non-linearity responses because the non-linear activation function skews the distribution of the pre-non-linearity responses (which are roughly normally distributed over the stimuli). Indeed, we find that Yeo-Johnson scaling noticeably increases the Pearson correlation (Fig. 6) with the pre-non-linearity responses for the noisy softplus models, almost as much as if you had directly applied the inverse of the softplus activation function to the post-non-linearity responses. We hypothesize that Yeo-Johnson scaling has a similar effect in the case of animal firing rates.

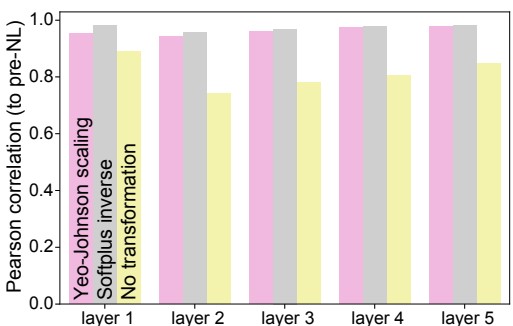

Figure 6: Correlation between post-non-linearity responses and pre-non-linearity responses after transforming the post-non-linearity responses in different ways (responses are for the noisy softplus models, averaged over 50 trials). We focus on correlation here because Yeo-Johnson scaling does not improve the $R^2$ score with respect to pre-non-linearity features (i.e. it does not directly match them), which makes sense as it is merely unskewing the distribution of post-NL features, which are already rather correlated with pre-NL features. Nevertheless, increased correlation implies that the pre-NL features can be more easily matched after linear re-weighting, as is done in Linear Nonlinear or Linear Softplus.

## F    IMPLEMENTATION DETAILS OF LINEAR NONLINEAR AND LINEAR SOFTPLUS

We implement Yeo-Johnson scaling with the PowerTransformer class in sklearn. The power transform fits one parameter. We put the PowerTransformer object followed by a *generalized linear model* (GLM) object into an sklearn Pipeline, so that the power parameter is only fit on the training data, not on test data.

The GLM object is created using the *glum* package. Each GLM specifies the inverse link function that relates the linear prediction to the response variable (such as ReLU, exponential or softplus), and the assumed noise structure in the response variable (Poisson noise in the case of Linear Nonlinear or Linear Softplus). The weights of the specified GLM are then optimized through Iterative Reweighted Least Squares.

The softplus inverse link function in LSP involves a scaling parameter $c$. When predicting noisy softplus model responses, we set $c = 100$, the same softplus scaling we used when training the models themselves. But when fitting LSP to predict mouse responses, we do not know *a priori* the optimal scaling parameter and must cross-validate values of $c$ along with the ridge penalty using GridSearchCV in sklearn.

