# OpenReview forum: "Inter-animal transforms as a guide to model-brain comparison"
_ICLR.cc/2024/Workshop/Re-Align — ICLR 2024 Workshop Re-Align ContributedTalk_

### Official Review · Reviewer_Qf6c · 2024-02-16
**Impressive work but a tough read at times, relevant workshop content, but missing some bits and pieces**

**Rating:** 2
**Fit:** 3
**Confidence:** 2

**Workshop Review:**

The authors examine model-brain alignment in a set of models aimed at linking DNN representations with firing rates from mouse data. The proposed transform that serves as the link between models and brains is the Inverse Linear Nonlinear Poisson model. The interesting goal here is to find transforms that respect the property of different layer properties while also being consistent across multiple models (or multiple models of different mice). DNN models serve as representatives of mice data initially and then used to assess inter-layer and inter-animal representations.

In some sense, talking about finding the linear mapping across models trained over different seeds and different data orderings in pre-ReLU responses feels kind of obvious, as you’re reasonably easily going to find this. However, the whole idea of inverting the nonlinearity for a ReLU, which zeros out negative values, is underspecified. Now, this point is addressed in the paper but I wasn’t really satisfied with the explanation that earlier layer unit correlations could be used to estimate a pre-ReLU response. I think something needs to be added to explain more about the mechanism of what it is you're getting at by inverting an activation and apply a different one and how this is consistent with measuring alignment between models and brains in order to gain useful insights into system behavior.

When models are trained under different seeds and training data orders, this was taken to be representative of the case of modelling multiple different mice in the real population. However, just training with a different seed and data order seems like your DNN population, representing the mice population, would be substantially more homogeneous than what you want find in nature. This equivalence did not sit so well with me as it was left unquantified and felt like quite a jump to make. One key concern rests on the ability of a transform to correctly map inter-animal, but it’s only with simulated spike counts from these quite similar models and I wondered if a stronger link could be made to verify this is a sound method.

When reaching Section 2, I found it difficult that no mention of the data input or task had been given at this point. We’ve already seen the nice picture of the neuron in Figure 2 and multiple instances of “trial-averaged” responses have been discussed. But, what’s the data? What’s the averaging over (figure talks about trial averaging)? I could not find any mention of the mouse data recording procedure or what is going on in this model. I am taking hints from mentions of the mouse "visual hierarchy" and the use of CNN models to infer this is an image task. I don't expect to rely so heavily on making my own inferences on important aspects of a paper when reviewing it. If the authors felt that the data / task / data partitions were irrelevant to the main goal of the paper, then that's something I have a bit of an issue with. What if the images being trained on exhibit a property that CNNs find specifically difficult (i.e. the texture / shape issue). I can only see that some models were made to match an ImageNet validation score and that the noisy softplus AlexNet models were made to match the ReLU-based AlexNet models. If there is an inference to be made that ImageNet is also the training data, that's one I'm not even sure about.

The culmination of underspecified fragments overall makes this quite tough to read. For example, just before the conclusion, we’re told about transform classes’ sensitivity to the different “functional signatures” but without any description of task, data, it removes any ability to mentally conceptualize how different brain areas might be functionally different for a “task”. What does this sentence mean? Some Figures have parts in that are not explained in the caption or text that left me a little confused.

Some sentences seem overly convoluted when they don’t need to be. There’s a beauty in formal writing, but I think you’re doing yourself a bit of a disservice by pushing so hard in that direction, at the expense of clarity in parts. The text is technically very well written and is precise (if found wanting, in some areas), but even as someone who knows a fair bit about this research, I found myself revisiting sentences sometimes over 5 times to make sure I picked up on all the intended nuances and trying to fill in the blanks (which a well-written paper shouldn’t force a reader to do).

From DNN similarity research, it’s known that many layers of a network in the heavily overparameterized regime are not doing many interesting things and I also wondered if you the paper could have included a few comments about this. It seems the fundamental basis lies in the fact that there is this link between brains and models, but the animal brain is a highly-optimized mechanism that likely does many important things during downstream processing of inputs. If a DNN model finds itself broadly self-similar among layers, is there a way to quantify this that would also show that inter-layer similarity breaks down for those specific layers under the suggested transforms being proposed? That would be interesting to know (this isn't a criticism, just a general thought during reading).

On the Figures:

Figure 1 is a good demonstration of the varies possibilities that can be encountered but I found it odd to see the (t1, t2, t3, … ) in part A. This isn’t defined anywhere else and I can’t see what it adds to the figure except potential confusion. The numbers aren’t linked to brain areas or colours so it’s just sort of floating there and would be more effective to be removed (if no explanation is given in caption / text, of course).

Figure 2 is a bit small. I think you can widen it and put some space between A and B.

I found Figure 6 quite difficult to understand. There’s a lot going on here and I’m not sure what the overall take-away message is without referring to the captions. I would consider finding a way to revise the figure in order to create a more salient visual “point”.  Even after 5 or 6 times of going back again, after 10 seconds I feel lost again.

Overall:

I really want to be constructive in my review and all points mentioned I hope are ones that authors can take into account to turn this promising paper into something even better, by removing some points of difficulty and increasing clarity, experimental details etc. I think if you had the scope for a longer paper, all the details would have been included that meant this would be an easier read. I get the sense you might have this longer work already, but needed to condense it down, but failed to adjust the text to compensate for the parts that were left out. So I have little doubt this is a very exciting piece of work, specifically in the area applied (animal / mouse models) but I also think you could have pivoted a bit more generally to how these ideas could be more widely of interest to the target audience of the workshop and pick up on a few recent intracranial studies in humans that could benefit from your analysis.

I will also state that animal models are not my area of expertise and the choices made for various modelling decisions are not choices I can evaluate sufficiently, so I could be missing great insight or overlooking other issues but I just want to be clear about that bit.

I think you should have given the draft to someone without any knowledge of the experiments and asked them to evaluate the paper because they would have been able to spot the same things I did and highlight some glaring omissions (mainly about the data set used) but also offer advise on being a bit vague in some wording and perhaps asking for more helpful links for why certain choices were made (i.e. new seed = model of a new mouse response).

**Reason For Not Giving Higher Score:**

I think the paper is going to be of interest to a fair few people at the workshop. It's a bit limited in terms of general conclusions, however, and restricted to modelling of neural firing rates. I think therefore it's a bit specialized (in addition to being speculative). So, as a poster, it could be something people visit out of interest to their own work and that I think would be the most natural place for this work.

**Reason For Not Giving Lower Score:**

I have a suspicion that issues I raised might stem from cutting down a longer piece of work to fit page limits and not adjusting the work enough for presentation in a long-format ICLR submission. I therefore think many of my issues arise from the inability to expand on explanations and hypotheses. I believe the authors will have great answers to all the questions, as my doubts do not come from doubting the ability of the authors to argue this point (at all). I think there is clear expertise and thought that has gone into this work that will hopefully turn out to be very promising.

Now, my inability to easily follow the paper I suspect is partially because it's out of my area of expertise (modelling firing rates in neurons from non-human animals) in addition to a lack of clarity on the authors' part for reasons given above.  However, I do expect there are people working in similar areas that will be able to extract much more insight from this work from discussions with the author(s).

**Reviewer Domain:**

cognitive science

---

### Official Review · Reviewer_YBy3 · 2024-02-23
**An extremely clear and well motivated paper on a particularly deep topic**

**Rating:** 3
**Fit:** 3
**Confidence:** 2

**Workshop Review:**

In this work the author’s ask how we should compare neural activity between models and brains. The author’s choose to focus on regression based methods as they provide a clear mapping between datasets (but they do also compare distance based methods). The authors then introduce two intuitive desiredata that any comparison metric should have. First, that it rates responses from the same area in different animals doing the same task highly and second that it rates the response between different brain areas lowly. These desiredata alone are already important contributions to the field, but the authors additionally go on to show that contrary to popular beliefs ridge regression is not flexible enough to show these desiredata. The Authors instead suggest that particular must be paid to the nonlinearity instantiated by the system. To test the authors derive two nonlinearities designed to mimic the biological nonlinearities of neurons and derive two functions for inverting those nonlinearities. The authors then show that performing regression using their new transform classes leads to increased area similarity while also increasing across area separability. Although the overall size of the effects is low, this work points to further improvements that could be made to increase effect size e.g considering a transform class that is bounded from above and below.

This work was particularly well written, and the claims made were strongly supported by the experiments.

**Reason For Not Giving Higher Score:**

N/A

**Reason For Not Giving Lower Score:**

This paper raises several interesting conceptual points that we should all consider as we seek to align models with brains. The only minor critique I had was that the overall effect size was rather small e.g the difference between basic ridge regression ( around 0.25) and the more complicated regression using the novel transform class was low ( around 0.27) when comparing inter-area similarity across mice. Despite, I think that the conceptual advances here are important enough to merit discussion in their own right.

**Reviewer Domain:**

neuroscience

---

### Official Review · Reviewer_mi4y · 2024-02-24
**Great work!**

**Rating:** 2
**Fit:** 3
**Confidence:** 3

**Workshop Review:**

This work tries to find a better transformation class to compare biological and artificial neural networks with. The authors go through multiple experiments, first on artificial, and then on biological neural networks to verify that their transformation class outperforms other common methods. Lastly, the authors conclude with results on an inter-mouse and artificial-to-biological dataset to verify their claims.

**Clarity** \
The paper is very clear overall, and I like how the authors very clearly explain their steps towards defining their final transformation class. However, I believe the clarity can be improved for a future full conference or journal submission.
- P5, in my own reading experience, I would have liked to have seen Figure 3 before the first results in Figure 2. I would then also suggest to move the second paragraph on P5 to an earlier part of the text. The authors dive into the results of a model they have not yet introduced, and this can be confusing.
- The authors first hypothesize on P3 “… that varying the random seed leads to model response variability for a given layer that is approximately similar to variability in animal responses for a given brain area.”, which is not substantiated by any references or explanation. The authors then go on to say on P4 that “This result highlights a hidden convergence between differently seeded model instances at intermediate layers. If no such convergence had occurred, that would have undermined our working hypothesis that the variability between model subjects is similar to variability between animal subjects, which are functionally similar for the same brain area.”. Although the authors do not directly conclude this, the authors do seem to suggest that their working hypothesis may be true given these results. I think their hypothesis is fine, but I do think there should be more of a substantiation for it, and the results in Figure 2 are not evidence of it in my opinion. Furthermore, I believe that even with a weaker statement, where the authors do not make a direct connection between different mice and seeds for a neural network, the reader can infer the importance of these results as the authors work towards a new class of transforms.
- I also think the authors should explain better why they first obtain the results for 2.1 and then move towards a “biologically motivated transform class” in Section 3. I think for a workshop it’s fine to include both analyses, but the second analysis clearly supersedes the first in terms of accuracy towards the authors’ main goal. I would suggest leaving out Section 2.1 in a full conference or journal paper.
- In all of the equations the authors provide, they should define the variables in their equation, e.g. Eq 1: what are X, A, Y, Eq 2 & 3: what are c, and A. I could infer these from the text, but I still believe it is good practice to define these variables nonetheless.

**Correctness** \
The analyses in the paper are correct as far as I can tell, and so are their derivations, and conclusions from their experiments. I would however implore the authors to release the code for their analyses upon acceptance, currently there is no mention of their code in the manuscript, except how they implement the models, and with what packages, which is definitely a good start! To improve the correctness for a potential full conference or journal paper I would suggest the following improvements.
- The authors mostly use the same-area similarity in terms of the R^2 score, which is a good measure of the similarity, but I believe (debiased CKA [1]) would be a good addition, since this a commonly used representational similarity measure with fewer invariances. Specifically, with R^2 the target is invariant under an invertible linear transform [1].
-  P2, In the caption of Figure 1, it says a given brain area is circled red, but I can’t find this area or circle.
- P3, “Our approach requires a good transform class to map accurately across subjects for the same brain area (while still distinguishing different brain areas), and a transform class that fails to do so will therefore be too strict. Moreover, it may turn out that a good transform class leads to similarity scores that do not satisfy metric criteria.”. I would also suggest that the authors include the uniqueness of the transform, if the solution to their optimization problem is not unique, how does this affect the results and how is a final solution selected?
- P8 “The fact that ILSP does not seem to perform significantly better…”. Please indicate in the text what significance test you performed and what the results are.
- P8, “We focus on LNP (instead of LSP) for computational efficiency, and, because we have direct access to the pre-non-linearity responses in our models…”. It would be interesting to see these results nonetheless to understand how well they align, this can be added to the Appendix.

[1] Kornblith, S., Norouzi, M., Lee, H., & Hinton, G. (2019, May). Similarity of neural network representations revisited. In International conference on machine learning (pp. 3519-3529). PMLR.

**Novelty** \
I think the paper is novel in that it adds a well thought-through method to the field that is based on biological neuron statistics and some interesting intuitions regarding pre and post-activation firing rates.

**Interest to the community** \
I believe this paper is of very direct interest to the community.

**Grammatical/spelling errors (these may be personal preference so feel free to ignore them)** \
- P5 “However, in practice, there may exist some function that approximately inverts ReLU by exploiting…” -> a ReLU

**Reason For Not Giving Higher Score:**

I believe the results and general conclusion of the paper are interesting and well-supported. The paper is also well-written, and I am sure the authors can develop this work further into a very insightful and great paper. However, I think the authors could increase the work’s clarity and reduce the repetitiveness of some of their results/claims.  Although I enjoy their walking through each step of the analysis, I also think they could have significantly shortened the paper by immediately focusing on their biologically motivated transform class. Furthermore, I believe the authors should spend more time on their inter-animal results, especially since it is so prominently displayed in their title and because their results (Figure 5c) are not as convincingly better on the actual mouse responses than they are for the artificial neural networks. Lastly, I think adding one or two more metrics to compare the models on would help, since what metric is best used for representational similarity analyses is a contentious topic in the field.

**Reason For Not Giving Lower Score:**

I think the paper is absolutely a good piece of work, and should be discussed at the workshop.

**Reviewer Domain:**

machine learning

---

### Decision · Program_Chairs · 2024-03-02

Accept (Contributed Talk)